# 12 new susceptibility loci for prostate cancer identified by genome-wide association study in Japanese population

Ryo Takata [ID] et al.[#]

Genome-wide association studies (GWAS) have identified ~170 genetic loci associated with prostate cancer (PCa) risk, but most of them were identified in European populations. We here performed a GWAS and replication study using a large Japanese cohort (9,906 cases and 83,943 male controls) to identify novel susceptibility loci associated with PCa risk. We found 12 novel loci for PCa including rs1125927 (*TMEM17*, $P = 3.95 \times 10^{-16}$), rs73862213 (*GATA2*, $P = 5.87 \times 10^{-23}$), rs77911174 (*ZMIZ1*, $P = 5.28 \times 10^{-20}$), and rs138708 (*SUN2*, $P = 1.13 \times 10^{-15}$), seven of which had crucially low minor allele frequency in European population. Furthermore, we stratified the polygenic risk for Japanese PCa patients by using 82 SNPs, which were significantly associated with Japanese PCa risk in our study, and found that early onset cases and cases with family history of PCa were enriched in the genetically high-risk population. Our study provides important insight into genetic mechanisms of PCa and facilitates PCa risk stratification in Japanese population.

Correspondence and requests for materials should be addressed to R.T. (email: rtakata@iwate-med.ac.jp) or to S.A. (email: akamats@kuhp.kyoto-u.ac.jp) or to H.N. (email: hidewaki@riken.jp). #A full list of authors and their affiliations appears at the end of the paper.

The incidence of prostate cancer (PCa) among Asian males has been dramatically rising even though it is still lower compared to that in western countries[1]. In 2015, PCa became the most common cancer type among Japanese males[2]. The best-known risk factor for PCa is family history[3]. A patient with a family history of PCa in first-degree relative has twice the risk of developing PCa during his lifetime compared to a patient without a family history, indicating a strong influence of inherited genetics on PCa susceptibility[3]. Early onset PCa is also recognized as a marker of genetic susceptibility for hereditary PCa, and PCa cases with rare variants of *BRCA2* and *HOXB13* constitute 2.0 and 3.1% of early onset cases[4,5]. To identify genetic polymorphisms associated with PCa, a number of genome-wide association studies (GWASs) have been conducted, which identified ~170 risk loci associated with PCa[6–10]. However, most of these studies were carried out for population of European ancestry or cohorts of mixed ethnicity. The considerable diversity of genetic background between different populations as shown by 1000 Genomes Project, together with the large difference in the incidence of PCa among ethnic groups, suggests a role of genetic risk factors in PCa disparities[11,12]. Even though a recent study suggests that the number of genetic loci associated with PCa has almost saturated after studying more than a hundred thousand patients[10], it is possible that even more PCa associated risk loci will be identified by studying populations of non-European ancestry.

We have initially identified five PCa associated risk loci by GWAS in the Japanese population[13], with a further meta-analysis in the Japanese population revealing three additional loci for PCa susceptibility[14]. We have also developed a highly reproducible polygenic risk estimation model for PCa detection, confirming polygenic risk of PCa in the Japanese population[15]. However, it is possible that increasing sample size will further improve power and may lead to the identification of yet-identified risk loci in this population. Moreover, it remains to be known which of the ~140 risk loci associated with PCa in the European ancestry population are also reproducible in the Asian population.

In this study, we have performed an even larger GWAS using an independent Japanese cohort to identify novel susceptibility loci associated with PCa and also validate the association of the previously reported risk loci with PCa risk in the Japanese population. Furthermore, we stratified the polygenic risk for Japanese PCa patients by using 82 significant single nucleotide polymorphisms (SNPs) and examined clinical feature of those at genetically high risk.

## Results

### GWAS in the Japanese population.
The overall design of the GWAS is depicted in Fig. 1. 5088 cases of PCa from Biobank Japan (BBJ) were included[16,17]. The controls consisted of 10,682 Japanese male subjects from four large cohort studies (the Japan Multi-Institutional Collaborative Cohort (J-MICC) study, the Japan Public Health Center-based Prospective (JPHC) study, Iwate Medical Megabank Organization (IMM), and Tohoku Medical Megabank Organization (ToMMo)) who had never been diagnosed with PCa[18–20]. Detailed clinical characteristics of the subjects are shown in Supplementary Table 1. After quality control, the association study was performed for 523,051 SNPs. All case and control samples belonged to the same cluster in the principal component analysis (Supplementary Fig. 1). A quantile–quantile (Q-Q) plot revealed modest inflation of the test statistics ($\lambda_{GC} = 1.189$); however, the value adjusted by the sample size, $\lambda_{GC}$ 1000, was 1.027 (Supplementary Fig. 2). We further conducted imputation analysis using the data of 275 Asians in the 1000 Genomes Project Phase 1 as a reference (JPT: 89, CHB: 97,

CHS: 89). As a result, 2997 SNPs represented $P < 1 \times 10^{-5}$ in GWAS (Fig. 1). Many of these SNPs existed in independent genetic regions including multiple loci at *8q24* (Fig. 2).

### Replication study and novel PCa-susceptibility loci.
Next, we conducted an independent replication study using 4818 cases from BBJ and JIKEI cohorts, and 73,261 male controls from the BBJ cohort. Of the 2997 SNPs, we removed SNPs that showed low imputation quality ($R$ square < 0.3) and SNPs that have previously been reported to be associated with PCa. Then we selected representative SNPs for each genetic region that showed the strongest association with PCa in the same linkage disequilibrium (LD) block. We also selected 20 SNPs which showed exceptionally strong association with PCa compared to the other SNPs in the same LD block. In total, we selected 101 SNPs for the replication study (Supplementary Table 2). In the replication study, genotyping was conducted using multi-index sequencing and the multiplex invader method (see Methods). Among the candidate SNPs, five SNPs could not be genotyped by either method and were excluded. When we combined both stages using the inverse method, 12 SNPs which have not been reported previously were identified to be significantly associated with PCa at $P < 5 \times 10^{-8}$ (Table 1).

Locus explore plot of each locus is shown in Fig. 3 and Supplementary Fig. 3[21]. All novel SNPs identified in this study except rs138708 existed in non-exonic regions. Of the 12 new loci, five loci (rs7542260 at chr.1, rs75777376 at chr.8, rs16901814 at chr.8, rs4554825 at chr.10, and rs8023793 at chr.15) contained no protein-coding genes in their LD blocks or in their vicinities (within 100 kb). On the other hand, rs11125927 at chr.2 was located at the intron of the *TMEM17* gene. rs73862213 at chr.3 was located near *GATA2*. rs77911174 at chr.10 region included *ZMIZ1* gene. rs11055034 at chr.12 was in the intron of *APOLD1* and the region also contained the 3' end of *CDKN1B*. rs6117562 at chr.20 was in the region containing *SLC52A3*. rs138708 at chr.22, formed a very large LD block with many genes, however, rs138708 is a non-synonymous SNP in the exon of *SUN2* gene. The region spanning rs4826594 at chr.X included three genes, *FGD1*, *GNL3L*, and *TSR2*.

### Expression analysis of the new loci.
It is possible that these SNPs reside in enhancer or suppressor regions of nearby genes and influence prostate carcinogenesis by altering the expression of these genes. In order to check the association between the genotype of newly identified SNPs and expression of the genes in their 1 Mb proximity, we conducted eQTL analysis using the GTEX database[22]. As a result, four SNPs showed weak association with expression of nearby genes ($P < 0.05$, Supplementary Table 3). Among them, rs16901814 showed association with expression of the *FAM84B* gene which is 311 kb away from the index SNP ($P = 0.0204$) (Supplementary Fig. 3b). Even though the function of *FAM84B* is not well known, its expression is elevated in various cancer, and its overexpression and copy number gain in PCa is reported to be associated with poor prognosis[23]. rs4554825 was associated with expression of *ZMIZ1-AS* ($P = 0.0361$). As previously mentioned, *ZMIZ1* is a gene that was present in the LD block containing rs77911174 (Fig. 3c). It is possible that both rs4554825 and rs77911174 are independently associated with PCa by altering expression or function of *ZMIZ1*. rs6117562 was associated with expression of *FAM110A* ($P = 0.0104$). *FAM110A* is a cell-cycle regulated gene whose function is not well known. Families of FAM110 proteins localize to centrosomes and are associated with microtubule aberrations[24], and *FAM110A* may affect prostate carcinogenesis through dysregulation of cell-cycle progression. rs6117562 was also associated

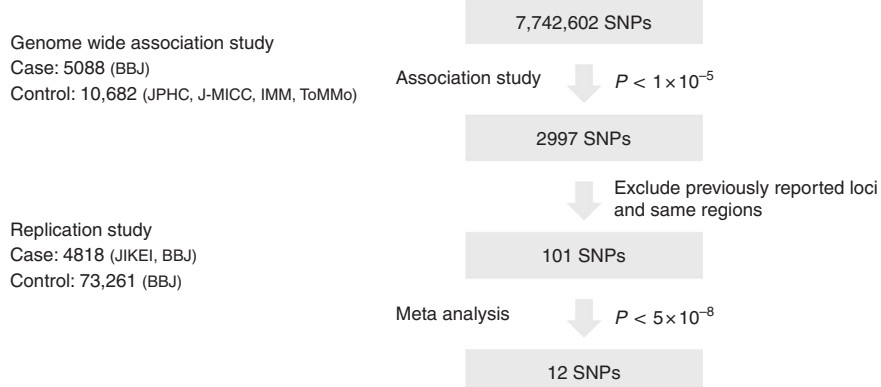

**Fig. 1** Scheme for study design and screening results. First, we conducted discovery GWAS followed by imputation using 1000 Genomes Project Phase1 as a reference. Of the 2997 SNPs that were associated with PCa ($P < 1 \times 10^{-5}$), top 101 SNPs excluding those residing in previously reported loci were evaluated in the replication study using an independent cohort. A total of 12 SNPs were significantly associated with PCa ($P < 5 \times 10^{-8}$) after meta-analysis of discovery and replication cohorts

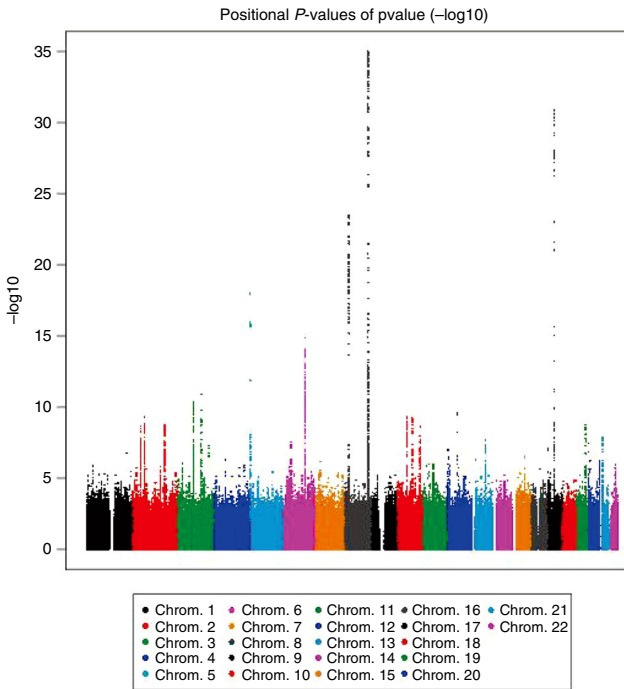

**Fig. 2** Manhattan plot of GWAS. $-\log10$ $P$ value is plotted on the $Y$-axis. Each $P$ value is calculated by a 1-degree-freedom Cochran-Armitage test

with *CSNK2A1* expression ($P = 0.0276$), which is known as responsible gene for Okur-Chung neurodevelopmental syndrome[25].

**Association with the reported loci**. Recent PCa GWAS have analyzed more than one-hundred thousand patients and identified many genetic regions associated with PCa[6–10]. However, these previous studies as well as meta-analysis have failed to identify the 12 susceptibility loci identified in this study (Supplementary Fig. 4). Among the novel loci identified, minor allele frequency (MAF) in Caucasians were significantly lower compared to that in the Japanese in seven loci, rs7542260, rs73862213, rs75777376, rs4554825, rs77911174, rs138708, and rs4826594 (Supplementary Table 4). The difference in MAF likely accounts for the lack of identification of these loci in previous studies. On

the other hand, the MAF in the European population did not significantly differ from that in the Japanese for the other SNPs. In these cases, LD structure pattern spanning the marker SNPs might be different by population.

Since several of the SNPs newly identified in this study existed in regions close to previously reported PCa susceptibility loci, we checked the independency of association by conditional analysis with GWAS data (Supplementary Table 5). rs721048, a known PCa susceptibility loci, is located near a newly identified SNP, rs11125927. However, rs11125927 was independently associated with PCa ($P = 1.02 \times 10^{-9}$) in the conditional analysis. We also checked for the independency between rs73862213 and *EEFSEC* region (rs10934853), and between rs4826594 and *NUT10/11* region (rs5945619); however, both rs73862213 and rs4826594 were statistically independent of the previously reported loci. Both rs75777376 and rs16901814 in 8q24 region, which exist near rs12543663, were also independently associated with PCa by conditional analysis. In addition, the two SNPs newly identified on chromosome 10 were independent of each other in the conditional analysis.

In the study, rs114780236 at chr.2 and rs4842687 at chr.12 showed significant association with Japanese PCa (Supplementary Table 6). But these SNPs were located in regions close to reported loci, rs58235267 and rs5799921[26]. Since our GWAS data did not contain the genotypes of rs58235267 and rs5799921, we could not conduct conditional analysis. Although these loci were not in complete LD with the reported loci in Japanese (The R square for rs114780236 and rs58235267 was 0.4701 and the R square for rs4842687 and rs579921 was 0.6962, respectively), the relatively high correlation suggests that rs114780236 and rs4842687 may be in the same susceptibility region with these reported loci in Japanese.

We examined whether the 167 previously reported PCa related loci are associated with PCa susceptibility in Japanese using our GWAS data. We excluded 13 SNPs for which data were not available and 18 SNPs which showed MAF < 0.01 or were mono-allelic, leaving 136 SNPs for analysis. We found 68 SNPs showed weak association ($P < 0.05$), and 28 SNPs to be strongly associated with PCa in Japanese after Bonferroni correction ($P < 0.00037$) (Supplementary Data 1). We then checked whether the rate of validation differs by the ethnic group the original report studied with 85 SNPs. The validation rate was the highest for the SNPs discovered using Asian cohort (Supplementary Table 7). Of the 75 SNPs found in whites, 34 (46%) were nominally significant at $P < 0.05$ in Japanese, with 18 at $P < 0.0005$. Of the 10 SNPs found

# ARTICLE

**Table 1 12 novel PCa-susceptibility loci identified by GWAS and replication study in Japanese population**

| | SNP ID | Chr | Position | Effect Allele | NonEffect Allele | | Case Frequency | Control Frequency | RSQR[a] | OR[b] | 95% CI | P-value | Nearby genes |
|---|---|---|---|---|---|---|---|---|---|---|---|---|---|
| 1 | rs7542260 | 1 | 5743196 | C | T | GWAS | 0.892 | 0.910 | 0.980 | 0.832 | (0.750–0.914) | $6.64 \times 10^{-06}$ | - |
| | | | | | | Replication | 0.887 | 0.902 | - | 0.850 | (0.784–0.916) | $1.14 \times 10^{-06}$ | |
| | | | | | | Meta | - | - | - | 0.843 | - | $\mathbf{3.99 \times 10^{-11}}$ | |
| 2 | rs11125927 | 2 | 62752975 | A | G | GWAS | 0.738 | 0.774 | 0.980 | 0.837 | (0.779–0.895) | $4.74 \times 10^{-10}$ | TMEM17 |
| | | | | | | Replication | 0.748 | 0.772 | - | 0.876 | (0.828–0.924) | $4.22 \times 10^{-08}$ | |
| | | | | | | Meta | - | - | - | 0.860 | - | $\mathbf{3.95 \times 10^{-16}}$ | |
| 3 | rs73862213 | 3 | 128217499 | A | G | GWAS | 0.908 | 0.929 | 0.996 | 0.754 | (0.666–0.842) | $1.71 \times 10^{-10}$ | GATA2 |
| | | | | | | Replication | 0.902 | 0.923 | - | 0.766 | (0.694–0.838) | $6.49 \times 10^{-14}$ | |
| | | | | | | Meta | - | - | - | 0.761 | - | $\mathbf{5.87 \times 10^{-23}}$ | |
| 4 | rs75777376 | 8 | 127796183 | T | C | GWAS | 0.896 | 0.908 | 0.528 | 0.779 | (0.669–0.889) | $6.08 \times 10^{-06}$ | - |
| | | | | | | Replication | 0.900 | 0.923 | - | 0.744 | (0.672–0.816) | $3.20 \times 10^{-16}$ | |
| | | | | | | Meta | - | - | - | 0.754 | - | $\mathbf{1.20 \times 10^{-20}}$ | |
| 5 | rs16901814 | 8 | 127881952 | G | A | GWAS | 0.854 | 0.833 | 0.993 | 1.176 | (1.108–1.244) | $2.00 \times 10^{-06}$ | - |
| | | | | | | Replication | 0.857 | 0.838 | - | 1.161 | (1.101–1.221) | $6.35 \times 10^{-07}$ | |
| | | | | | | Meta | - | - | - | 1.167 | - | $\mathbf{5.64 \times 10^{-12}}$ | |
| 6 | rs4554825 | 10 | 80244623 | C | T | GWAS | 0.790 | 0.764 | 1.000 | 1.170 | (1.110–1.230) | $1.06 \times 10^{-07}$ | ZMIZ1-AS |
| | | | | | | Replication | 0.786 | 0.766 | - | 1.122 | (1.070–1.174) | $6.29 \times 10^{-06}$ | |
| | | | | | | Meta | - | - | - | 1.142 | - | $\mathbf{8.43 \times 10^{-12}}$ | |
| 7 | rs77911174 | 10 | 80826833 | A | G | GWAS | 0.604 | 0.643 | 0.957 | 0.853 | (0.801–0.905) | $6.14 \times 10^{-10}$ | ZMIZ1 |
| | | | | | | Replication | 0.599 | 0.634 | - | 0.864 | (0.820–0.908) | $8.68 \times 10^{-12}$ | |
| | | | | | | Meta | - | - | - | 0.859 | - | $\mathbf{5.28 \times 10^{-20}}$ | |
| 8 | rs11055034 | 12 | 12890626 | C | A | GWAS | 0.601 | 0.577 | 0.999 | 1.123 | (1.073–1.173) | $3.11 \times 10^{-06}$ | CDKN1B, APOLD1 |
| | | | | | | Replication | 0.606 | 0.580 | - | 1.115 | (1.073–1.157) | $3.74 \times 10^{-07}$ | |
| | | | | | | Meta | - | - | - | 1.119 | - | $\mathbf{6.05 \times 10^{-12}}$ | |
| 9 | rs8023793 | 15 | 66942093 | A | C | GWAS | 0.595 | 0.559 | 0.997 | 1.137 | (1.087–1.187) | $2.56 \times 10^{-07}$ | - |
| | | | | | | Replication | 0.592 | 0.569 | - | 1.098 | (1.056–1.140) | $1.27 \times 10^{-05}$ | |
| | | | | | | Meta | - | - | - | 1.114 | - | $\mathbf{2.81 \times 10^{-11}}$ | |
| 10 | rs6117562 | 20 | 753310 | G | A | GWAS | 0.485 | 0.516 | 0.989 | 0.874 | (0.824–0.924) | $3.64 \times 10^{-08}$ | SLC2A3, FAM110A |
| | | | | | | Replication | 0.498 | 0.512 | - | 0.945 | (0.903–0.987) | $7.12 \times 10^{-03}$ | |
| | | | | | | Meta | - | - | - | 0.915 | - | $\mathbf{3.34 \times 10^{-08}}$ | |
| 11 | rs138708 | 22 | 39138332 | G | A | GWAS | 0.855 | 0.831 | 0.983 | 1.172 | (1.104–1.240) | $3.74 \times 10^{-06}$ | SUN2 |
| | | | | | | Replication | 0.859 | 0.834 | - | 1.218 | (1.158–1.278) | $5.12 \times 10^{-11}$ | |
| | | | | | | Meta | - | - | - | 1.198 | - | $\mathbf{1.13 \times 10^{-15}}$ | |
| 12 | rs4826594 | X | 54454406 | G | A | GWAS | 0.482 | 0.525 | 0.967 | 0.924 | (0.888–0.960) | $8.21 \times 10^{-06}$ | FGD1, GNL3L, TSR2 |
| | | | | | | Replication | 0.479 | 0.508 | - | 0.943 | (0.913–0.973) | $1.08 \times 10^{-04}$ | |
| | | | | | | Meta | - | - | - | 0.935 | - | $\mathbf{7.13 \times 10^{-09}}$ | |

[a]RSQR, imputation accuracy. SNPs were imputed in the GWAS
[b]Non-effect alleles were considered as reference

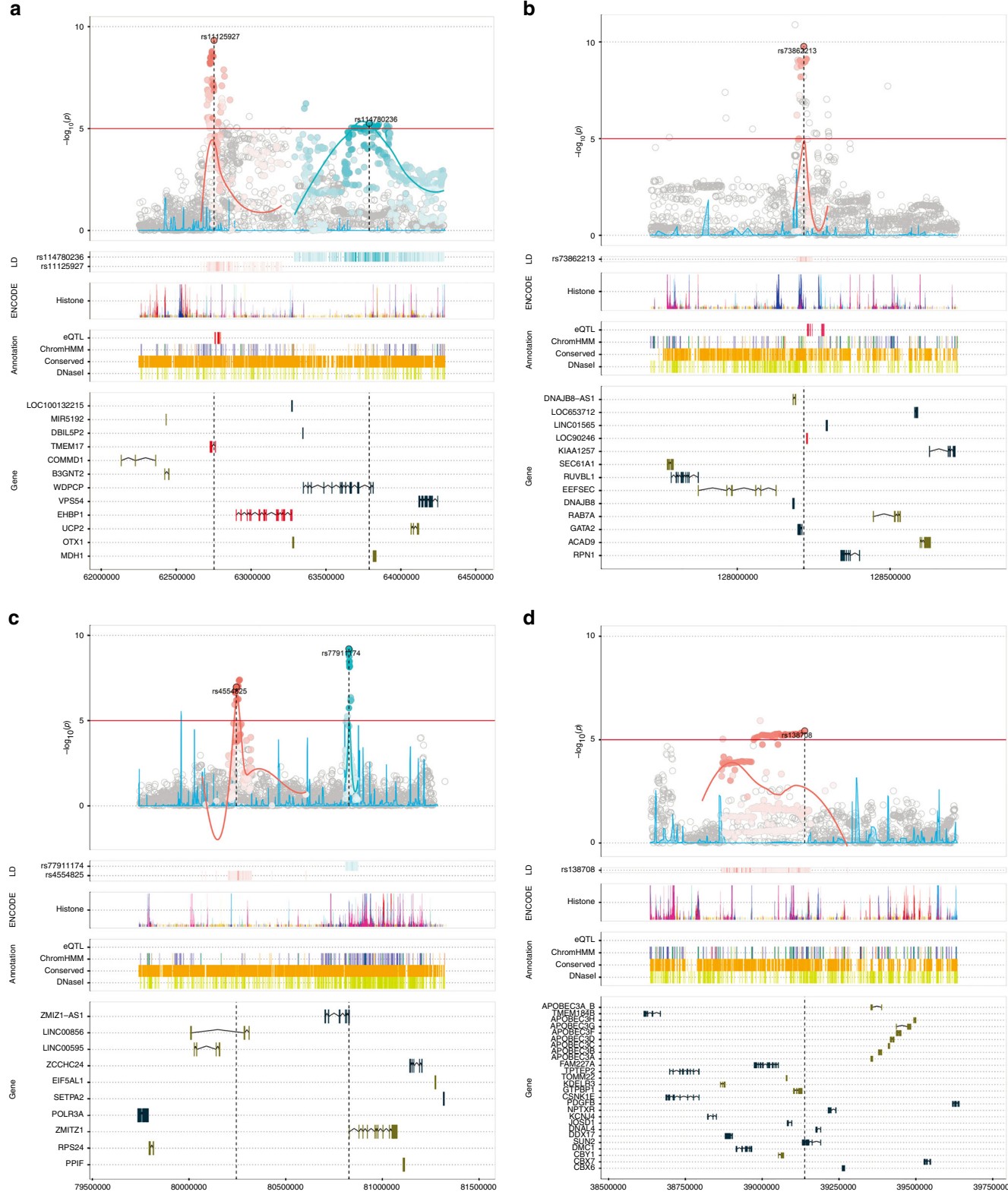

previously in Asian men, 8 (80%) were nominally significant ($P <$ 0.05) with all at $P < 0.0005$. The result highlights the large heterogeneity of PCa associated genetics factors between different ethnic groups.

**Polygenic risk score in Japanese population**. Finally, we selected 12 SNPs which were newly discovered in the study and 68 SNPs

which showed nominal association with the Japanese PCa risk from previously reported SNPs (Supplementary Table 8). Since our GWAS data did not contain two reported loci, rs58235267 and rs5799921, we also included rs114780236 and rs4842687 to represent the two regions. We calculated a polygenic risk score (PRS) by counting the number of risk alleles and their effects in each individual of the GWAS samples. The distribution of the PRS in the PCa cases ($n = 4893$) and the male controls ($n =$

**Fig. 3** Locus Explorer plots of novel five GWAS loci. **a** rs11125927 at chr.2. **b** rs73862213 at chr.3. **c** rs4554825 & rs77911174 at chr.10. **d** rs138708 at chr.22. The regional association plot ($-\log10(P)$ panel) depicts variant $P$-values relative to chromosomal position. Variants in linkage disequilibrium with the novel lead SNP(s) at $r^2 \geq 0.1$ according to the 1000 Genomes JPT population are shaded in the Manhattan plot and linkage disequilibrium track (LD panel), with darker color denoting stronger correlation with the lead variant. Lower sections of the plot indicate the relative positions of genes and selected biological annotations. Annotations displayed are: histone modifications in ENCODE tier 1 cell lines (Histone track), the positions of variants that are eQTLs with prostate tumor expression in TCGA prostate adenocarcinoma samples (eQTL track), chromatin state categorizations in the PrEC cell-line by ChromHMM (ChromHMM track), the position of conserved element peaks (Conserved track) and the position of DNaseI hypersensitivity site peaks in ENCODE prostate cell lines (DNaseI track). Genes on the positive and negative strand are denoted by brown and turquoise color respectively (Gene track). The horizontal axis represents genomic coordinates in the hg19 reference genome

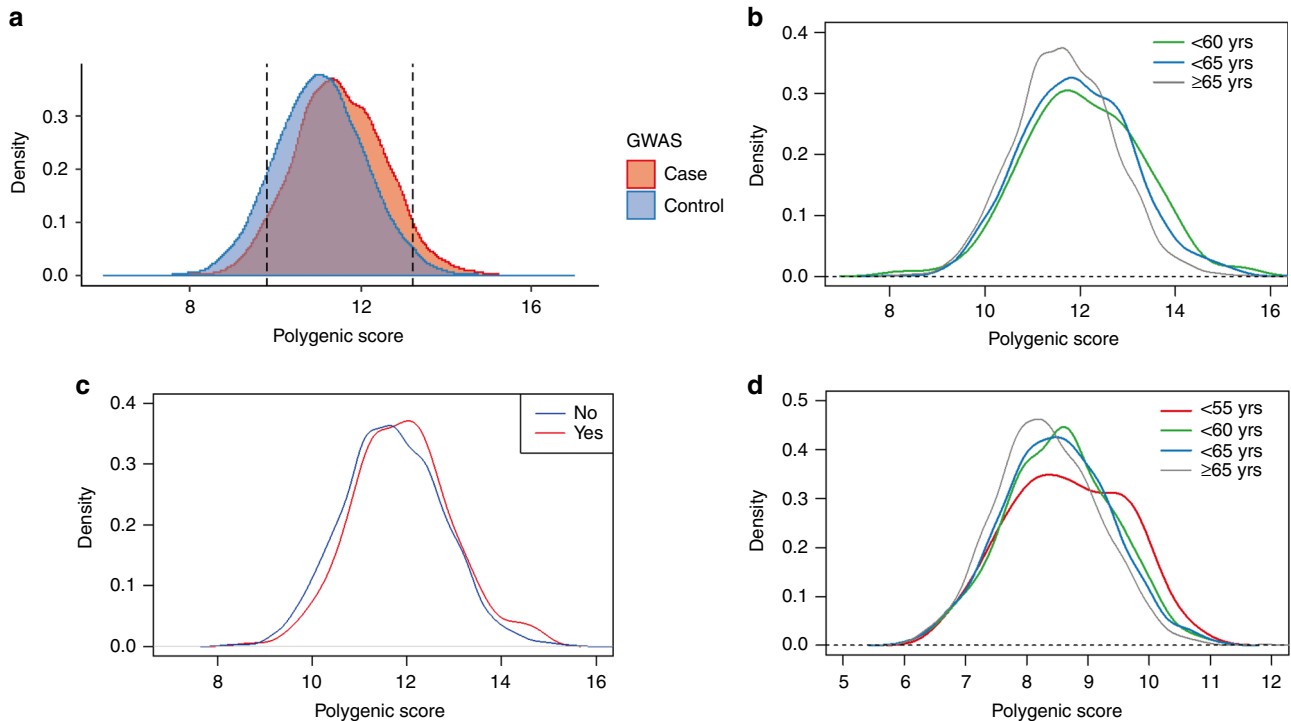

**Fig. 4** The distribution of the polygenic risk score (PRS) for PCa in Japanese population. **a** The PRS distribution of the PCa cases ($n = 4893$) and the male control ($n = 10,682$) of GWAS. Density was estimated using the Gaussian kernel. The 5% higher and lower percentiles are shown as dotted lines. **b** The PRS distribution by the age at diagnosis of PCa in the GWAS cases ($n = 4762$). Density was estimated using the Gaussian kernel. Green, younger than 60 years ($n = 129$); blue, younger than 65 years ($n = 781$); gray, 65 years or older ($n = 3852$). **c** The PRS distribution by the presence of PCa family history in the GWAS cases ($n = 4893$). Density was estimated using the Gaussian kernel. Red, positive PCa family history ($n = 272$); blue, negative PCa family history ($n = 4621$). **d** The PRS distribution by the age at diagnosis of PCa in the JIKEI validation cohort ($n = 2218$). Density was estimated using the Gaussian kernel. Red, younger than 55 years ($n = 94$); green, younger than 60 years ($n = 310$); blue, younger than 65 years ($n = 802$); gray, 65 years or older ($n = 802$)

10,682) are shown in Fig. 4a. We defined the upper 5% of the cases ($n = 245$) as the genetic high-risk group and the lower 5% as the low-risk group ($n = 245$), and examined clinical features of these genetically risk–stratified groups. Notably, we found that the mean diagnosis age of the high-risk group was 2.7 years younger than the non-high risk group (mean age 68.7-year old vs 71.4 year old, $P = 6.54 \times 10^{-8}$, by t-test), while we observed little difference of mean diagnosis age between the low-risk and non-low risk groups (72.3 vs 71.1, $P = 0.020$). We observed the enrichment of high PRS in early onset PCa cases ($P = 0.00221$ for cases with age < 60-year old, by t-test, and $P = 4.30 \times 10^{-9}$ for cases with age < 65-year old, Fig. 4b). The high-risk group was also enriched with patients who have a positive family history of PCa ($P = 0.00339$, by Fisher test, Fig. 4c). On the other hand, when we recalculated the PRS using 150 SNPs after adding 68

reported SNPs which indicated no association with Japanese PCa, statistical association between PRS and early onset PCa in the high-risk group became weaker ($P = 0.02395$ for cases with age < 60-year old, by t-test, and $P = 3.24 \times 10^{-7}$ for cases with age < 65-year old, Supplementary Fig. 5a). Association between PRS and positive family history of PCa also declined ($P = 0.02395$, by Fisher test, Supplementary Fig. 5b).

Furthermore, we calculated PRS in the PCa cases of the two validation cohorts (BBJ $n = 2386$, and JIKEI $n = 2218$) by counting the risk allele of 63 SNPs for which genotypes were available in the replication study (Supplementary Table 8). The distribution of PRS in both cohorts is shown in Supplementary Fig. 5c. We confirmed that in both cohorts, the age at diagnosis for the 5%-high-risk group is approximately 2 years younger than the non-high risk group ($P = 0.023$ in BBJ cohort and $P = 0.0061$

in JIKEI cohort). We observed that early onset PCa was enriched in the high PRS group of the BBJ cohort ($P = 0.0111$ for cases with age < 60-year old, by t-test, and $P = 0.0451$ for cases with age < 65-year old, Supplementary Fig. 5d) and in the JIKEI cohort ($P = 0.00378$, for cases with age < 55-year old, $P = 0.00119$ for cases with age < 60-year old, and $P = 8.90 \times 10^{-5}$ for cases with age < 65-year old, by t-test, Fig. 4d). We also confirmed that the high-risk group was enriched with patients who have positive family history of PCa in BBJ replication cohort ($P = 0.0281$, by Fisher test, Supplementary Fig. 5e).

## Discussions

In this study, we identified 12 new PCa-susceptibility loci in the Japanese population, but their functional or biological significances related to PCa development are still unclear, which is also the case with many other previously reported loci. Because most of them are located in non-coding regions and our expression analysis found only four loci to be weakly related to expression of nearby genes, these loci are likely associated with gene regulatory functions.

Among the 12 loci, rs11125927 at chr.2 contained *TMEM17* gene in the LD block, which is a cilium associated gene reported to suppress invasion and migration of non-small cell lung cancer by restoring *Occuludin* and *Zo-1* expression through inactivation of ERK-P90RSK-Snail pathway[27]. rs73862213 at chr.3 contained *GATA2* in the same LD block, which plays an important role in promoting high grade PCa[28]. In addition to its role as a transcription factor that promotes androgen receptor (*AR*) binding and activation, it regulates a subset of clinically relevant PCa associated genes in an AR independent manner. Functionally, *GATA2* overexpression promotes cell motility, migration, growth, tumorigenesis, and therapy resistance in PCa. rs77911174 at chr.10 region included the *ZMIZ1* gene, which binds to AR and enhances its transcriptional activity in PCa cells[29]. In addition, it co-localizes with AR and SUMO1 and promotes sumoylation of AR in vivo. The same genetic region has previously been reported to be associated with susceptibility to colon cancer and breast cancer[30,31]. Our eQTL analysis suggested the association of *ZMIZ1-AS* (antisense) expression with another independent loci rs4554825, indicating that regulation of *ZMIZ1* expression should be important in prostate carcinogenesis. The LD block spanning rs11055034 at chr.12 contained the 3' end of *CDKN1B*, which is strongly expressed in non-proliferating cells and plays important roles in the regulation of both quiescence and G1 progression[32]. It is known to act as a tumor-suppressor in PCa and suppression of *CDKN1B* leads to growth promotion of PCa cells[32]. rs6117562 at chr.20 was in the region containing *SLC52A3*, which is known as a transporter for riboflavin, however, association with PCa has not been reported[33]. rs138708 at chr.22 is a non-synonymous SNP of *SUN2*, which is a member of LINK complex and plays an important role in nuclear-cytoplasmic connection and suppress Warburg effect in cancer[34]. Overexpression of *SUN2* inhibits PCa cell growth and *SUN2* knockdown promotes PCa growth[35]. The LD block spanning rs4826594 at chr.X included three genes, *FGD1*, *GNL3L*, and *TSR2*. *Fgd1* is transiently associated with invadopodia and required for their formation and function in extracellular matrix degradation and acts by direct modulation of Cdc42 activation[36]. *FGD1* is overexpressed in human prostate and breast cancer and is associated with tumor aggressiveness[36]. *GNL3L* has been reported to directly bind to and stabilize MDM2 protein, however its role in PCa has not been studied[37]. *TSR2* is known to enhance apoptosis by suppressing NF-kB signaling in laryngeal cancer[38]. The genes listed above potentially influence PCa development and further functional analysis is warranted.

Polygenic risk estimation by using common variants for PCa has been attempted. For breast cancer, PRS consisting of 77 SNPs tested in 33,000 cases demonstrated significant interaction between PRS and age and family history[39]. Genetic risk prediction of PCa was first reported using five common susceptibility variants[40], which was established by simply counting the number of risk alleles. Subsequently, models incorporated increasing number of the significant variants[15], and some models uses much more variants including the variants that are not significant at the genome-wide level[41]. It is still controversial which and how many variants should be used for PRS[42]. Ethnic-specific or genomic structure-specific information is also an important issue for PRS. In this study for PRS we used 82 SNPs which were statistically associated with PCa in Japanese population and found that early onset and familial PCa were enriched in the high-risk group in Japanese population. Early onset and familial PCa, with which genetic factors should be more involved, was reported to be enriched in cases with rare variants of *BRCA2* and *HOXB13* in Caucasian population[4,5]. However, this is the first report to show that risk assessment using 50~ common variants may explain hereditary phenotype of PCa. In addition, PRS using 82 SNPs which are associated with Japanese PCa showed stronger association with early onset PCa and positive family history of PCa than PRS with 150 SNPs which also included the SNPs that failed to show association with Japanese PCa. The result suggests that more precise selection of patients with high risk of developing PCa may be possible with ethnic population-specific PRS. Early onset PCa is often more aggressive and may have a different etiology than later-onset PCa[43]. Among men diagnosed with high grade and advanced stage PCa, men with early onset PCa are more likely to die of their cancer, with higher cause-specific mortality than later-onset disease[43]. In addition, familial PCa accounts for a greater proportion of PCa in early onset cases than it does in men diagnosed at older ages. These PCas have been shown to have a more significant genetic component indicating that this group may benefit the most from evaluation of genetic risk[44]. Since the PRS model is useful for the early detection of early onset PCa and familial PCa, PRS might have a greater impact on the clinical examination and treatment of PCa. However, further replication of this risk-stratification by other larger cohort is required before the model is applied to clinical use.

As with other GWAS for PCa, contamination of control with undiagnosed PCa is a limitation of our study. In Japan, as with western countries, increasing number of PCa are detected by PSA screening. However, it is estimated that still less than 50% of males over 50 are actually exposed to PSA screening in Japan, and 10% of newly diagnosed PCa patients present with metastatic disease. Therefore, it is likely that the control in this study includes undiagnosed PCa cases to certain extent. This certainly implies that the SNPs identified in this study is potentially associated with factors other than PCa carcinogenesis, such as health consciousness to receive screening. Continuing efforts should be made to reveal the biological significance of each SNPs reported in GWAS studies including this study, which hopefully delineate the complex interaction between genetic susceptibility and environmental exposure.

In summary, we have conducted a large-scale GWAS for Japanese and identified 12 PCa susceptibility loci that have not been reported previously. We stratified the polygenic risk for Japanese PCa patients by using 82 associated-SNPs and indicated that early onset and familial PCa cases were enriched in the genetically high-risk population.

## Methods

**Study population**. GWAS included 5088 cases from BioBank Japan, which was established in the Institute of Medical Science at the University of Tokyo[16,17]. Among the 5088 cases for the GWAS, 272 (5.3%) subjects had a family history of PCa, 1219 (23.9%) subjects revealed PSA ≥ 10 and 1989 (39.1%) subjects were diagnosed to have cancer with Gleason score of 7 or higher (Supplementary Table 1). From the BBJ, pathologically proven PCa cases were selected. Non-cancer controls were from three population-based cohorts, including the J-MICC study[18], the JPHCStudy[19], IMM, and ToMMo[20]. Genomic DNA samples were extracted from peripheral blood leukocytes and normal tissues using a standard method. All participating studies obtained informed consent from all participants by following the protocols approved by their institutional ethical committees before enrollment, and the ethical committees at each institute approved the project (BBJ: https://biobankjp.org/english/index.html, J-MICC: http://www.jmicc.com/en/, JPHC: https://epi.ncc.go.jp/en/jphc/index.html, IMM: http://iwate-megabank.org/en/, ToMMo: https://www.megabank.tohoku.ac.jp/english/).

**Genotyping and quality control**. GWAS was conducted using Illumina OmniExpress Exome or the OmniExpress + HumanExome BeadChip (Illumina Inc., San Diego, California, U.S.). Of the 947,830 SNPs genotyped, 195,588 were mono-allelic and were excluded from further analysis. Cluster plots of the top 100 SNPs showing the smallest $P$-values were checked by visual observation, and 604,992 SNPs met the criteria of call rate ≧ 0.99 both in case and control samples. Finally, SNPs $P ≧ 1.00 \times 10^{-6}$ in a Hardy-Weinberg Equilibrium test were selected. Association study was performed for the total of 523,051 SNPs.

Imputation of the un-genotyped SNPs was conducted by MaCH[45] and minimac[46] using the data from the JPT/CHS/CHD subjects and using the 1000 Genome Project Phase 1 (release 16 March 2012) as a reference. We exclude SNP with a large allele frequency difference between the reference panel and the GWAS (>0.16)[47]. We also excluded SNPs with low imputation quality score (R square < 0.3) and insertion/deletion polymorphisms.

**Samples and genotyping for the replication studies**. We conducted a replication study using independent 4818 PCa cases and 73,261 controls. Case samples were obtained from the BioBank Japan (2236 cases) and JIKEI samples (2582 samples) from The Jikei University School of Medicine. The JIKEI sample has been described previously with new samples being added for this study[48]. All cases were histologically diagnosed by local pathologists, and clinical data were collected by local urologists. Controls in the replication study were the 73,261 male samples from BBJ that were subjected to GWAS for diseases other than PCa.

A multi-index PCR-based target sequencing method was used to sequence the target region of case samples[49]. We used a two-step PCR method to construct DNA libraries. The 1st PCR (25 cycle) was performed with 202 primer pairs and 2X Platinum Multiplex PCR Master Mix (Thermo Fisher Scientific) to amplify the target region, followed by the 2nd PCR (4 cycle) with 8-bp barcode and adapter sequences added using primers targeting shared 5′ overhangs introduced during the 1st PCR and KAPA HiFi HotStart DNA Polymerase (KAPA). After purification and quantification of pooled libraries, we sequenced them by 2 150-bp paired-end reads on a HiSeq 2500 (Illumina) instrument. Sequence reads allocated to each individual were aligned to the human reference sequence (hg19) using Burrows-Wheeler Aligner (ver. 0.7.12) and processed using Genome Analysis Toolkit (GATK, ver. 3.4–46)[50,51]. For quality control, we selected individuals in which more than 98% of the target region was covered with 20 or more sequencing reads. We called variants of each individual separately using UnifiedGenotyper and HaplotypeCaller of GATK, and VCMM (ver. 1.0.2)[52]. Genotypes for all individuals were jointly determined for each variant based on the sequencing read ratio of reference and alternative alleles. When the alternative allele frequency was between 0 and 0.15, between 0.25 and 0.75, and between 0.85 and 1, we assigned homozygote of the reference allele, heterozygote, and homozygote of the alternative allele, respectively. The SNPs that could not be analyzed by multi-index sequencing were genotyped by multiplex PCR-based invader assay[53]. The five SNPs that could not be genotyped by both assays were excluded in the replication study.

**Statistical analysis**. In all stages, association of each SNP was assessed under an additive model. In the GWAS, the genetic inflation factor $\lambda_{GC}$ was derived from $P$-values obtained by the Cochran-Armitage trend test for all the tested SNPs. The quantile–quantile plot was drawn using the R program. $\lambda_{GC\ 1000}$ was calculated using the following formula:[54]

$$\lambda_{GC\ 1000} = 1 + (1 - \lambda_{GC\ obs}) \times (1/n_{cases} + 1/n_{controls})/(1/1000_{caes} + 1/1000_{controls}).$$

Odds ratios were calculated using the non-effect alleles as references, unless stated otherwise. The results of the combined analyses of the GWAS and the replication study were verified by the Mantel–Haenszel method. Heterogeneity across the two stages was examined using P-link[55]. We considered $P = 5 \times 10^{-8}$ (GWAS and meta-analysis) as the significance threshold after Bonferroni correction for multiple testing.

**Polygenic risk score**. Polygenic score was computed as the weighted sum of the number of risk alleles. Log odds ratios computed in the GWAS part of this study were used as the weights. The number of incorporated SNPs was 82 and 63 in the GWAS and the validation cohorts, respectively (Supplementary Table 8). For imputed alleles in GWAS, dosage values were used as the number of risk alleles. Statistical association between polygenic score and clinical information was analyzed using the statistical software R[56].

**Reporting summary**. Further information on research design is available in the Nature Research Reporting Summary linked to this article.

## Data availability

GWAS summary statistics of prostate cancer will be publicly available at our website (JENGER, http://jenger.riken.jp/en/) and the National Bioscience Database Center (NBDC, https://humandbs.biosciencedbc.jp/en/) Human Database. Genotype data of case samples are available at NBDC under research ID hum0014.

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

## Acknowledgements

The authors acknowledge the staff of the BBJ Project, the JPHC Study, the J-MICC Study, IMM and ToMMo, for their outstanding assistance in collecting samples and clinical information. The BioBank Japan (BBJ) and Japanese GWAS were supported by the Ministry of Education, Culture, Sports, Sciences and Technology of the Japanese government. The J-MICC Study was supported by Grants-in-Aid for Scientific Research for Priority Areas of Cancer (No. 17015018) and Innovative Areas (No. 221S0001) and by JSPS KAKENHI Grants (No. 16H06277) from the Japanese Ministry of Education, Culture, Sports, Science and Technology. The JPHC Study has been supported by the National Cancer Center Research and Development Fund since 2011 and was supported by a Grant-in-Aid for Cancer Research from the Ministry of Health, Labour and Welfare of Japan from 1989 to 2010.

## Author contributions

R.T., S.A., and H.N. directed and designed the study and wrote the manuscript. R.T., A.T., M.F., Y.K., and S.A. performed statistical analysis. Y.M., K.A., K.N., and H.N. performed genotyping in replication study. E.S. and C.H. analyzed the new loci. H.Y., Y.N., A.H., K.M., K.W., T.Y., N.S., M.I., S.T., M.S., A.S., K.T., N.M., K.S., K.M., M.K., S.E., O.O., W.O., and H.N. contributed to sample and data acquisition. J.I., O.O., and W.O. acquired the funding.

## Additional information

**Competing interests:** The authors declare no competing interests.

Ryo Takata [1,2], Atsushi Takahashi [3,4], Masashi Fujita [2], Yukihide Momozawa[5], Edward J. Saunders[6], Hiroki Yamada[7], Kazuhiro Maejima[2], Kaoru Nakano[2], Yuichiro Nishida [8], Asahi Hishida[9], Keitaro Matsuo [10,11], Kenji Wakai[9], Taiki Yamaji[12], Norie Sawada [12], Motoki Iwasaki[12], Shoichiro Tsugane [13], Makoto Sasaki [14], Atsushi Shimizu [14], Kozo Tanno[14], Naoko Minegishi[15], Kichiya Suzuki [15], Koichi Matsuda [16], Michiaki Kubo[17], Johji Inazawa[18], Shin Egawa[7], Christopher A. Haiman[19], Osamu Ogawa[20], Wataru Obara[1], Yoichiro Kamatani [3], Shusuke Akamatsu[2,20] & Hidewaki Nakagawa[2]

[1]Department of Urology, Iwate Medical University, Morioka 020-8505, Japan. [2]Laboratory for Cancer Genomics, RIKEN Center for Integrative Medical Sciences, Yokohama 230-0045, Japan. [3]Laboratory for Statistical Analysis, RIKEN Center for Integrative Medical Sciences, Yokohama 230-0045, Japan. [4]Department of Genomic Medicine, National Cerebral and Cardiovascular Center Research Institute, Suita 564-8565, Japan. [5]Laboratory for Genotyping Development, RIKEN Center for Integrative Medical Sciences, Yokohama 230-0045, Japan. [6]The Institute of Cancer Research, London SW7 3RP, UK. [7]Department of Urology, Jikei University School of Medicine, 105-8461 Tokyo, Japan. [8]Department of Preventive Medicine, Faculty of Medicine, Saga University, 840-8502 Saga, Japan. [9]Department of Preventive Medicine, Nagoya University Graduate School of Medicine, Nagoya 466-8550, Japan. [10]Division of Cancer Epidemiology and Prevention, Aichi Cancer Center Research Institute, 464-8681 Nagoya, Japan. [11]Department of Cancer Epidemiology, Nagoya University Graduate School of Medicine, Nagoya 466-8550, Japan. [12]Division of Epidemiology, Center for Public Health Sciences, National Cancer Center, Tokyo 104-0045, Japan. [13]Center for Public Health Sciences, National Cancer Center, 104-0045 Tokyo, Japan. [14]Iwate Tohoku Medical Megabank Organization, Iwate Medical University, Yahaba 028-3694, Japan. [15]Tohoku Medical Megabank Organization, Tohoku University, Sendai 980-8573, Japan. [16]Laboratory of Clinical Genome Sequencing, Department of Computational Biology and Medical Sciences, Graduate School of Frontier Sciences, The University of Tokyo, 108-8639 Tokyo, Japan. [17]RIKEN Center for Integrative Medical Sciences, Yokohama 230-0045, Japan. [18]Department of Molecular Cytogenetics, Medical Research Institute, Tokyo Medical and Dental University, Tokyo 113-8510, Japan. [19]Center for Genetic Epidemiology, Department of Preventive Medicine, Keck School of Medicine, University of Southern California, Los Angeles, California 90033, USA. [20]Department of Urology, Kyoto University Graduate School of Medicine, Kyoto 606-8501, Japan

