## [Peer Review File · Nature Communications]

Reviewers' comments:

Reviewer #1 (Remarks to the Author):

This paper by Takata et al is an important piece of work, advancing our understanding of the complex underlying architecture of prostate cancer susceptibility. The discovery is important but needs to be done more carefully- especially in mapping against known regions identified in the larger PRACTICAL studies.

The PRS is useful to assess and should be further compared to the CEU PRS. It is surprising that they did not look at correlates in the known hits not 'replicated in the Japanese' to enhance and improve the PRS. This should be done for comparison as it is unlikely that there are that many ancestral-specific variants. The issue is one really of alleles frequency and LD measurements.

There is a major issue that must be addressed- the claim of novelty is not quite accurate. The current paper is somewhat loose and has made some clear mistakes- and there could be more.

For the 14 'new' loci, 2 are already known.

1. rs4842687 is definitely not a novel signal as it correlates with rs5799921 (in Schumacher Nat Genet paper, phase3 indel, missing in Riken) with a n LD of $r^2=0.99$ in CEU, $r^2=0.53$ in EAS

2. rs114780236 looks like it is related to rs58235267(UK FM 99) $r^2 = 0.088$ (phase3 EUR), $r^2=0.27$ (phase3 EAS)

Lets be sure the other 12 are distinctive and novel.....

Reviewer #2 (Remarks to the Author):

The paper contains findings from a genome-wide association study (GWAS) of prostate cancer risk amongst ethnic Japanese. A previous prostate cancer GWAS for Japanese was reported by this group in 2010. The current study interrogates a new, larger cohort of Japanese men. The primary study

population featured 5,088 prostate cancer cases from BioBank Japan at the University of Tokyo. Whether these cases were identified as a consequence of screening or of symptomatic disease presentation was not clear. The 10,682 control subjects were from three population-based cohorts. How the absence of prostate cancer or absence of prostate cancer risk was determined for these controls was also not described. A second validation population study featured 4,818 prostate cancer cases and 73,261 control subjects from BioBank Japan.

The study itself identified some 14 genetic loci, largely in non-coding regions of the genome, where single nucleotide polymorphisms (SNPs) were associated with prostate cancer. Only 4 of these SNPs appeared to be associated with gene regulation. The SNPs were distinct from those discovered thus far in GWAS analyses of prostate cancer in European men. In addition, a polygenic risk model for Japanese prostate cancer was constructed using 82 SNPs that was over-represented among early onset and familial prostate cancer.

Prostate cancer in ethnic Japanese men potentially provides interesting insights into the etiology of prostate cancer for all men. As mentioned in the paper, the disease burden is increasing in Japan (and throughout much of Asia). Ethnic Japanese men have been known to adopt higher prostate cancer risk when immigrating to North America. Despite these hints at environmental contributions to prostate cancer development in North American (and increasingly in Japan), heredity is well-known to play a significant role in prostate carcinogenesis for men with European ancestry. The current paper illustrating prostate cancer risk associations throughout the genome for Japanese men buttresses this theme. Presumably, a significant fraction of genetic risk alleles will be associated with gene-environment interactions, perhaps underscoring inter-individual differences in cell and genome damage and repair.

As for the data presented, the genotyping and GWAS methods appeared to be solidly performed. The major concerns for the study and its conclusions focus on definitions of cases and of controls. In North America, most prostate cancer is never diagnosed; thus, a diagnosis of prostate cancer tends to reflect adoption of screening than disease biology per se. This thwarts many attempts at genetic analyses. Large population-based GWAS where prostate screening is common tend to deliver associations with serum prostate-specific antigen (PSA) levels (which can be affected by androgen action, inflammatory processes, benign prostate enlargement, etc.) and general health-seeking behaviors. Familial studies are corrupted by the tendency of brothers of men with prostate cancer to be more likely to get screened, and as a result, to be diagnosed at earlier ages. Also, many control subjects are undiagnosed cases. To overcome these limitations, some prostate cancer geneticists have focused on high-grade prostate cancer or on lethal prostate cancer for risk association studies.

To address these biases, the authors should provide significantly more detail on the age distribution of cases/controls, the number of cases diagnosed by screening, etc.

Reviewer #1:

This paper by Takata et al is an important piece of work, advancing our understanding of the complex underlying architecture of prostate cancer susceptibility. The discovery is important but needs to be done more carefully- especially in mapping against known regions identified in the larger PRACTICAL studies. The PRS is useful to assess and should be further compared to the CEU PRS. It is surprising that they did not look at correlates in the known hits not 'replicated in the Japanese' to enhance and improve the PRS. This should be done for comparison as it is unlikely that there are that many ancestral-specific variants. The issue is one really of alleles frequency and LD measurements.

REPLY:

Thank you very much for your thoughtful suggestion. The author agrees with the reviewer that it is important to compare the PRS incorporating only the SNPs validated in the Japanese and the PRS which includes all the previously reported PCa associated SNPs. To answer the question, we have calculated PRS with 150 SNPs which included all known hits regardless of the association with Japanese PCa. Interestingly, statistical association between PRS and high-risk group or early-onset PCa were weaker with PRS incorporating 150 SNPs compared to the PRS using only 82 Japanese PCa associated SNPs, implying the importance of population-specific PRS. Therefore, we added the description, "On the other hand, when we recalculated the PRS using 150 SNPs after adding 68 reported SNPs which indicated no association with Japanese PCa, statistical association between PRS and early-onset PCa in the high-risk group became weaker ($P=0.02395$ for cases with age<60 year-old, by t-test, and $P=3.24 \times 10^{-7}$ for cases with age<65 year-old, Supplementary Figure 5a). Association between PRS and positive family-history of PCa also declined ($P=0.02395$, by Fisher test, Supplementary Figure 5b)." in the result section, and "In addition, PRS using 82 SNPs which are associated with Japanese PCa showed stronger association with early-onset PCa and positive family history of PCa than PRS with 150 SNPs which also included the SNPs that failed to show association with Japanese PCa. The result suggests that more precise selection of patients with high risk of developing PCa may be possible with ethnic population-specific PRS." in the discussion.

There is a major issue that must be addressed- the claim of novelty is not quite accurate. The current paper is somewhat loose and has made some clear mistakes- and there could be more.

For the 14 'new' loci, 2 are already known.

1. rs4842687 is definitely not a novel signal as it correlates with rs5799921 (in Schumacher Nat Genet paper, phase3 indel, missing in Riken) with a n LD of $r^2=0.99$ in CEU, $r^2=0.53$ in EAS

2. rs114780236 looks like it is related to rs58235267(UK FM 99) $r^2 = 0.088$ (phase3 EUR), $r^2=0.27$ (phase3 EAS)

Lets be sure the other 12 are distinctive and novel.....

REPLY:

For the newly identified SNPs, we have tried to check the association between the known nearby signals as much as possible by conditional analysis. However, since our GWAS data did not contain rs58235267 and rs5799921, we could not conduct conditional analysis and we could not clarify independence of these SNPs. Therefore, we excluded the two SNPs from newly identified SNPs, and we described the issue in results as “In the study, rs114780236 at chr.2 and rs4842687 at chr.12 showed strong association with Japanese PCa (Supplementary Table 6). These SNPs were located in regions close to reported loci, rs58235267 and rs579992126). Since our GWAS data did not contain rs58235267 and rs5799921, we could not conduct conditional analysis. However, for the rs4842687, even though R square for the two SNPs was 1.0 in the Caucasians, it was 0.6962 in Japanese and 0.5337 in East Asian, which suggests the possibility that rs4842687 is an independent PCa susceptible loci only in East Asian including Japanese.” The title of the manuscript has also been changed accordingly.

Reviewer #2:

The paper contains findings from a genome-wide association study (GWAS) of prostate cancer risk amongst ethnic Japanese. A previous prostate cancer GWAS for Japanese was reported by this group in 2010. The current study interrogates a new, larger cohort of Japanese men. The primary study population featured 5,088 prostate cancer cases from BioBank Japan at the University of Tokyo. Whether these cases were identified as a consequence of screening or of symptomatic disease presentation was not clear. The 10,682 control subjects were from three population-based cohorts. How the absence of prostate cancer or absence of prostate cancer risk was determined for these controls was also not described. A second validation population study featured 4,818 prostate cancer cases and 73,261 control subjects from BioBank Japan.

The study itself identified some 14 genetic loci, largely in non-coding regions of the genome, where single nucleotide polymorphisms (SNPs) were associated with prostate cancer. Only 4 of these SNPs appeared to be associated with gene regulation. The SNPs were distinct from those discovered thus far in GWAS

analyses of prostate cancer in European men. In addition, a polygenic risk model for Japanese prostate cancer was constructed using 82 SNPs that was over-represented among early onset and familial prostate cancer.

Prostate cancer in ethnic Japanese men potentially provides interesting insights into the etiology of prostate cancer for all men. As mentioned in the paper, the disease burden is increasing in Japan (and throughout much of Asia). Ethnic Japanese men have been known to adopt higher prostate cancer risk when immigrating to North America. Despite these hints at environmental contributions to prostate cancer development in North American (and increasingly in Japan), heredity is well-known to play a significant role in prostate carcinogenesis for men with European ancestry. The current paper illustrating prostate cancer risk associations throughout the genome for Japanese men buttresses this theme. Presumably, a significant fraction of genetic risk alleles will be associated with gene-environment interactions, perhaps underscoring inter-individual differences in cell and genome damage and repair.

As for the data presented, the genotyping and GWAS methods appeared to be solidly performed. The major concerns for the study and its conclusions focus on definitions of cases and of controls. In North America, most prostate cancer is never diagnosed; thus, a diagnosis of prostate cancer tends to reflect adoption of screening than disease biology per se. This thwarts many attempts at genetic analyses. Large population-based GWAS where prostate screening is common tend to deliver associations with serum prostate-specific antigen (PSA) levels (which can be affected by androgen action, inflammatory processes, benign prostate enlargement, etc.) and general health-seeking behaviors. Familial studies are corrupted by the tendency of brothers of men with prostate cancer to be more likely to get screened, and as a result, to be diagnosed at earlier ages. Also, many control subjects are undiagnosed cases. To overcome these limitations, some prostate cancer geneticists have focused on high-grade prostate cancer or on lethal prostate cancer for risk association studies.

To address these biases, the authors should provide significantly more detail on the age distribution of cases/controls, the number of cases diagnosed by screening, etc.

REPLY

Thank you very much for the important comments regarding the general limitation of GWAS in PCa. According to the comments by the reviewer, we have provided detailed characteristics of each subject in Supplementary Table 1.

In the GWAS, age of control subjects was younger than that of cases. Therefore, the control may include

patients who are at risk that have not yet developed PCa. In addition, since still less than 50% of males over 50 are exposed to PSA screening in Japan, the control may be contaminated by undiagnosed PCa patients. However, since inclusion of such subjects makes the difference between the cases and controls smaller, the findings in the study is still considered to be robust. As the reviewer correctly mentioned, there is certain concern that the SNPs identified in the present study is potentially associated with factors other than PCa carcinogenesis, such as health consciousness to receive screening, which also applies to other GWAS studies.

We have added following comments in Discussion. "As with other GWAS for PCa, contamination of control with undiagnosed PCa is a limitation of our study. In Japan, as with western countries, increasing number of PCa are detected by PSA screening. However, it is estimated that still less than 50% of males over 50 are actually exposed to PSA screening in Japan, and 10% of newly diagnosed PCa patients present with metastatic disease. Therefore, it is likely that the control in the present study includes undiagnosed PCa cases to certain extent. This certainly implies that the SNPs identified in the present study is potentially associated with factors other than PCa carcinogenesis, such as health consciousness to receive screening. Continuing efforts should be made to reveal the biological significance of each SNPs reported in GWAS studies including the present study, which hopefully delineate the complex interaction between genetic susceptibility and environmental exposure."

REVIEWERS' COMMENTS:

Reviewer #1 (Remarks to the Author):

the authors have revised the manuscript suitably in response to the reviewers' comments. The new title and more nuanced discussion of the 12 novel findings plus 2 more represents an improvement.

there is one place that needs some further editing- the comments on the two known regions. The r^2 for the Japanese and East Asians are still quite high (both above 0.5) so this reviewer does not agree that these explain a possible new signal. between distinct MAFs and r^2 , it is still likely these are the same- and should be fully acknowledged as such with minor editing. as written, it is misleading and not really correct....

Reviewer #2 (Remarks to the Author):

The authors have largely addressed my questions.

Reviewer #1 (Remarks to the Author):

the authors have revised the manuscript suitably in response to the reviewers' comments.

The new title and more nuanced discussion of the 12 novel findings plus 2 more represents an improvement.

there is one place that needs some further editing- the comments on the two known regions. The r^2 for the Japanese and East Asians are still quite high (both above 0.5) so this reviewer does not agree that these explain a possible new signal. between distinct MAfs and r^2 , it is still likely these are the same- and should be fully acknowledged as such with minor editing. as written, it is misleading and not really correct....

REPLY:

Thank you very much for providing the comment.

The author agrees with the reviewer comment that two SNPs might be in the same lesions of known hits. To answer the remarks, we have revised the result to reflect reviewers comment as “Although these loci were not in complete linkage disequilibrium with the reported loci in Japanese (The R square for rs114780236 and rs58235267 was 0.4701 and the R square for rs4842687 and rs579921 was 0.6962 respectively), the relatively high correlation suggests that rs114780236 and rs4842687 may be in the same susceptibility region with these reported loci in Japanese.”. In addition, we replaced the title of the supplementary table 6 as “Japanese PCa susceptibility SNPs that might be in the same lesions of reported loci.”.